# VIVIDPORTRAITS: FACE PARSING GUIDED PORTRAIT ANIMATION

## ABSTRACT

Portrait animation aims to transfer the facial expressions and movements of a target character onto a reference character. This task presents two main challenges: accurately transferring motion and expressions while fully preserving the identity features of the reference portrait. We introduce Vividportraits, a diffusion-based model designed to effectively meet these objectives. In contrast to existing methods that rely on sparse representations such as facial landmarks, our approach leverages facial parsing maps for motion guidance, enabling a more precise conveyance of subtle expressions. A random scaling technique is applied during training to prevent the model from internalizing identity-specific features from the driving images. Furthermore, we perform foreground-background segmentation on the reference portrait to reduce data redundancy. The long-video generation process is refined to improve consistency across sequences. Our model, exclusively trained on public datasets, demonstrates superior performance relative to current state-of-the-art methods, achieving a notable 8% improvement in expression metric. More visual results are available on the anonymous website https://www.vividportraits.cn.

## 1 INTRODUCTION

The objective of the portrait animation task is to transfer facial expressions and movements from a target video to a reference image, allowing the reference character to accurately replicate the facial dynamics of the target character (Ma et al., 2024; Xie et al., 2024). Typically, these two characters are distinct individuals. This field presents two primary challenges: the full preservation of the identity features of the reference portrait, including crucial aspects such as facial structure, texture, and individual characteristics, and the accurate transfer of motion expressions. The latter involves not only the replication of larger, more noticeable movements but also the precise conveyance of subtle facial expressions, such as minor changes in the eyes or mouth, which are essential for generating natural and believable animations. Portrait animation technology has been extensively utilized in various domains, including video conferencing (Khakhulin et al., 2022), live streaming (Qu et al., 2023), and e-commerce (Sun et al., 2023). This task is typically achieved using GAN-based models or diffusion-based models.

GAN-based models (Siarohin et al., 2019b; Drobyshev et al., 2022; Liu et al., 2023; Wang et al., 2021; Siarohin et al., 2019a) initially derive a motion "flow field" to simulate the movement of features within the feature map of the reference image. The feature map, distorted by this flow field, is subsequently refined via a decoder and further optimized through GAN training. However, these models exhibit two fundamental limitations: (1) The motion "flow field" is capable of simulating alterations in the existing features of the reference image but has difficulty representing absent or occluded features. (2) GANs suffer from mode collapse issues, preventing them from guiding the decoder to achieve flawless image generation. Consequently, GAN-based models frequently produce outputs characterized by blurriness, unrealistic artifacts, and implausible content.

Diffusion models have been effectively employed in portrait animation tasks (Chang et al., 2023; He et al., 2023; Hu, 2024; Xu et al., 2023; 2024b; Zeng et al., 2023; He et al., 2024; Ma et al., 2024; Xu et al., 2024a; Zhang et al., 2023a). Some approaches (Hu, 2024; Ma et al., 2024) employ landmarks to represent facial poses and expressions; however, this sparse representation inevitably results in the loss of crucial motion information, such as the actions of the eyebrows and mouth,

failing to adequately capture subtle movements. Other techniques (Xie et al., 2024; He et al., 2024; Yang et al., 2024) circumvent intermediate motion representations, directly using the target image to guide the generation process. RGB inputs introduces an excess of identity features from the target portrait, causing a blend of reference and target portraits and introducing unrelated background noise, which leads to noticeable artifacts in the generated output. Additionally, some methods (He et al., 2023; Qu et al., 2023) utilize facial parameters to construct motion representations, but these approaches overly depend on the accuracy of the facial encoder.

In this paper, we introduce Vividportraits, an innovative portrait animation method grounded in diffusion models. To address the issues encountered in prior models, a suite of effective techniques is proposed to enhance the quality of portrait animation: (1) To achieve a more precise and accurate extraction of motion information, we propose facial parsing maps, a region-based motion representation. Compared to traditional point-based facial landmark methods, facial parsing maps preserve the complete structure of facial expressions, allowing for more accurate conveyance of subtle facial expressions. (2) A "random scaling" technique was introduced for the facial parsing maps to prevent the incorporation of identity features from the target portrait. The processed parsing maps retain motion information while modifying identity features, thereby enabling the model to be trained effectively under cross-id scenarios. (3) A foreground-background separation strategy is introduced to ensure each model component receives the necessary information, avoiding interference from redundant noise. (4) We strengthened the long-video generation method by integrating the "cyclic overlap" technique, enhancing consistency between video frames by increasing the focus on each individual frame.

In summary, our contributions are as follows:

- We propose a novel diffusion-based, zero-shot portrait animation framework that enables precise control over facial poses and expressions while preserving the character's identity.

- A new motion representation is introduced, coupled with a cross-id training method, which accurately captures expression and motion features while mitigating the impact of appearance-related characteristics.

- A foreground-background separation method is presented, enhancing the extraction of relevant information, reducing redundancy, and improving the interpretability of the framework.

- The long-video generation strategy is optimized, further enhancing temporal consistency between video frames.

## 2 RELATED WORK

### 2.1 GAN-BASED METHODS

Portrait animation is a specialized image generation task that has evolved in tandem with the advancements in generative models, particularly GANs (Goodfellow et al., 2020) and diffusion models (Song et al., 2020a; Ho et al., 2020). Numerous GAN-based approaches address portrait animation by decomposing the task into three distinct stages (Xie et al., 2024): the first stage involves encoding the reference image to obtain its feature map, thereby capturing the identity characteristics of the reference; the second stage generates a flow field from the reference and target images to simulate the displacement of feature points within the reference image; and the third stage utilizes this flow field to manipulate the feature map and decodes it to produce the final animated result.

Various optimizations have been introduced in the generation of the flow field. Some methods utilize sparse keypoints (Siarohin et al., 2019b; Hong & Xu, 2023; Wang et al., 2021; Zhao & Zhang, 2022; Geng et al., 2018) to model the flow field, as exemplified by FOMM (Siarohin et al., 2019b), which trains a keypoint detector to capture the relative movement of facial features. Utilizing keypoints and relative displacement as the flow field can mitigate identity leakage between different individuals; however, it necessitates that the pose of the person in the first frame of the target video closely resembles that of the reference.

Furthermore, some approaches forgo keypoints altogether and adopt alternative strategies for generating the flow field, such as leveraging depth maps (Hong et al., 2022), pose and expression parame-

ters (Ren et al., 2021; Qu et al., 2023; Sun et al., 2023; Drobyshev et al., 2022; Qu et al., 2023), and tri-planes (Siarohin et al., 2021). These methods endeavor to optimize flow field generation from various perspectives, thereby enhancing the quality of portrait animation. However, they remain constrained by the flow field, which continues to impose stringent requirements on the initial pose of the target character. If there is a significant discrepancy between the pose of the reference portrait and the initial pose of the target portrait, the generated results are likely to be subpar.

## 2.2 DIFFUSION-BASED METHODS

Diffusion models (Song et al., 2020b; Ho et al., 2020; Song et al., 2020a) have achieved significant success in various tasks, including image generation (Gu et al., 2024; Song et al., 2020b; Saharia et al., 2022; Ruiz et al., 2023), image editing (Ye et al., 2023; Cao et al., 2023), and video editing (Liu et al., 2024; Ma et al., 2023; Qi et al., 2023) applications. Consequently, many studies have begun applying diffusion models to portrait animation. Mainstream diffusion-based portrait animation frameworks harness the robust generative capabilities of Stable Diffusion (Rombach et al., 2022) and integrate temporal modules (Guo et al., 2023) to enhance inter-frame consistency.

There are multiple approaches for representing facial expression movements, including the direct use of portrait images (Xie et al., 2024; Yang et al., 2024), facial encoding (He et al., 2023; Xu et al., 2023), and facial landmarks (Hu, 2024; Ma et al., 2024).

Utilizing portrait images introduces background elements and other information unrelated to motion, while also transmitting redundant identity information—such as facial structure and size—from the target portrait to the model, thereby complicating the training process.

Facial encoding employs an encoder to represent facial movements, with the accuracy of the encoder being a critical determinant of the method's effectiveness. However, as a simplified feature representation, facial encoding often fails to capture subtle variations and complex expressions of the face, resulting in information loss. Additionally, this approach is sensitive to noise, which can lead to instability in the generated output.

Facial landmarks are another commonly employed method for motion representation. However, the number of keypoints significantly influences the precision of expression representation, with common counts being 5, 68, 81, and 98. These sparse, point-based representations overlook substantial facial detail, resulting in incomplete expression transfer and hindering the accurate conveyance of subtle facial movements.

## 3 PRELIMINARIES

Diffusion models generate images by iteratively denoising a sample drawn from Gaussian noise $z_T \sim \mathcal{N}(\mathbf{0}, \mathbf{1})$ over $T$ steps. The Latent Diffusion Model (Blattmann et al., 2023), a crucial component of Stable Diffusion (Rombach et al., 2022), executes the denoising process within the latent space of a pretrained autoencoder, which significantly enhances the efficiency and stability of the training process. More specifically, a Variational Autoencoder (Kingma, 2013) maps real images from the RGB space to a lower-dimensional latent space.

Within this latent space, a UNet (Ronneberger et al., 2015) carries out the denoising task under the influence of text conditions. This is facilitated through the use of self-attention and cross-attention mechanisms embedded within Transformer blocks. The text conditions are injected into the UNet via cross-attention, thereby steering the denoising process. The training objective of the LDM is defined as follows:

$$\mathbb{E}_{z_0, t, \epsilon \sim \mathcal{N}(\mathbf{0}, \mathbf{1})} \left[ \left\| \epsilon - \epsilon_\theta \left( z_t, t \right) \right\|_2^2 \right] \tag{1}$$

where $\epsilon$ represents the ground truth noise at time step $t$, and $\theta$ encompasses the trainable parameters of the UNet.

Several models introduce improvements to Stable Diffusion (SD) to facilitate portrait or full-body animation tasks. These models typically encompass the following key components: (1) ReferenceNet (Hu, 2024): A network that mirrors the architecture of SD's UNet, which processes the

reference image encoded by the VAE (Kingma, 2013) to extract identity and background information. This extracted information is subsequently injected into the denoising UNet's self-attention layers to guide the generation process. (2) Temporal Attention (Guo et al., 2023): This module executes self-attention across frames within the denoising UNet to preserve temporal consistency during video generation, thereby ensuring that the resulting animation is smooth and coherent. (3) Control Module (Zhang et al., 2023b): Commonly realized as a ControlNet, this module incorporates motion information by blending action images (such as skeleton maps and depth maps) with noise. This integration is critical for accurately simulating the movements and poses of the target character. (4) CLIP (Radford et al., 2021): Utilized to encode the reference image, replacing traditional text conditions, and embedding this encoded information into both the ReferenceNet and Denoising UNet to more closely align the generated output with the characteristics of the reference image.

## 4 METHOD

As depicted in the Figure 1, we employ a structure akin to that of preceding works, but introduce optimizations in motion and identity information to more accurately represent facial poses and expressions while ensuring identity consistency. VAE (Kingma, 2013) is used as the encoder and decoder and CLIP (Radford et al., 2021) image encoder is incorporated as a replacement for the traditional text encoder. We employ a ReferenceNet, identical in structure to the DenoisingUNet but without the temporal layer, to encode identity information. Additionally, a pose encoder similar to the one utilized in AnimateAnyone (Hu, 2024) is employed.

Subsequently, we will elaborate on our random scaling method for facial parsing maps in Section 4.1. In Section 4.2, the foreground-background separation mechanism is detailed. Lastly, Section 4.3 introduces the optimized strategy for long-video generation.

### 4.1 RANDOM SCALING OF FACIAL PARSING MAPS

The critical elements of the portrait animation task are expression transfer and identity preservation, with the representation of facial expressions being a pivotal step in achieving effective expression transfer. An optimal motion representation can accurately convey the demeanor, expressions, and even the emotional nuances of the target portrait, resulting in more realistic and expressive generated outcomes.

To overcome the limitations of the aforementioned methods, we propose adopting facial parsing maps as a motion representation. We employ FARL (Zheng et al., 2022) to extract the facial parsing map from the target portrait, removing expression-irrelevant components and augmenting it with the missing ocular motion information. This approach effectively eliminates background elements unrelated to motion, retaining solely the components pertinent to facial expressions and thereby circumventing the introduction of extraneous information. Moreover, the facial parsing map preserves the complete shape of each facial component, ensuring the full transmission of expression and motion information, avoiding the information loss associated with sparse point-based representations.

As mentioned above, the current facial parsing maps lack representation of the eyeball region, and relying on them alone would lead to the omission of essential eye movement information. To mitigate this limitation, we utilize the keypoint detection technique from FARL (Zheng et al., 2022) to delineate the keypoints of the eye region, including the eyeball position, in a manner consistent with the facial parsing map. These keypoints are then incorporated into the final target parsing map. This approach ensures that the model captures the essential expression information related to the relative position of the eyeball. Given that both the facial parsing map and landmarks are derived using methods from FARL, their alignment is inherently maintained.

To attenuate the influence of identity-related attributes such as size and spacing in cross-id scenarios, we introduce a random scaling technique for the facial parsing maps. Initially, we eliminate the contours in the facial parsing map, preserving only elements such as the eyes and nose, thereby removing facial shape information from the map. In the subsequent step, each component is randomly scaled individually within a range of 0.8 to 1.2, modifying the size of each element within the parsing map. In the final step, the entire face is scaled within a range of 0.7 to 1.3, adjusting the spacing between the components in the parsing map.

Throughout the overall scaling process, we maintain the position of the face within the complete image and preserve the relative positions of each facial component within the face. This methodology allows us to generate target facial parsing maps that exhibit varied identity characteristics while keeping the expressions and poses unchanged. By utilizing these parsing maps, which alter identity features while retaining motion features, cross-id training is facilitated with precise ground truth from the outset. This ensures that the model is dedicated to learning expression and motion information from the parsing maps, effectively disentangling identity-related specifics.

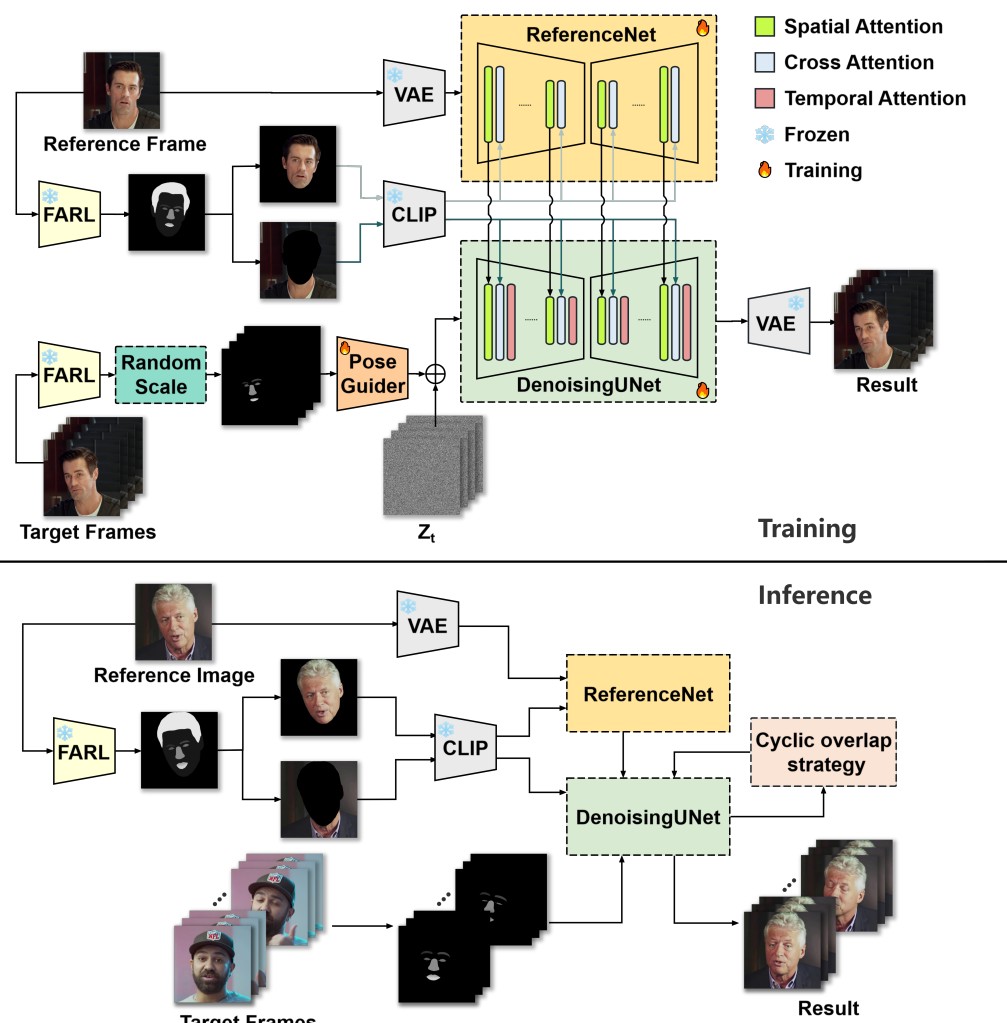

Figure 1: **Overview of Vividportraits.** During the training phase, we utilize FARL to extract the facial parsing map of the target portrait, which is then processed using our random scaling technique. The processed features are extracted by the PoseGuider and subsequently fused with multi-frame noise. The reference portrait is segmented into foreground and background based on its facial parsing map, with each part being integrated through ReferenceNet and DenoisingUNet, respectively. During the inference phase, the facial parsing map of the target portrait is not subjected to random scaling; however, cyclic overlap is applied after each denoising step to ensure consistency.

## 4.2 FOREGROUND-BACKGROUND SEPARATION MECHANISM

In previous works, CLIP (Radford et al., 2021) is predominantly used to process the entire reference portrait image, serving as the prompt for both ReferenceNet and DenoisingUNet. Consequently, both networks receive information pertaining to the identity features of the reference portrait as

well as the background. We posit that ReferenceNet should primarily focus on extracting identity features. Therefore, the facial segmentation map of the reference image is employed to partition it into a foreground, containing identity features, and a background that is unrelated to identity.

ReferenceNet is provided exclusively with the foreground identity information encoded by CLIP, which enables the Transformer blocks within it to place greater emphasis on identity features during the attention operation. Given that the prompt contains only identity information without interference from background elements, ReferenceNet can concentrate on encoding identity features while allocating minimal attention to background information.

The reduction in background information is compensated for by DenoisingUNet. We use CLIP to encode the background portion of the reference image and pass this encoding to DenoisingUNet instead of the entire image. This approach allows DenoisingUNet to receive a background encoding from CLIP that is devoid of identity information contamination, thereby supplementing the background information that is reduced by our foreground-background separation mechanism in ReferenceNet.

As demonstrated in the Appendix, our foreground-background separation mechanism facilitates improved preservation of both identity and background features in the generated results.

### 4.3 LONG VIDEO GENERATION STRATEGY

Typically, diffusion-based portrait animation methods generate a fixed number of frames in a single iteration, such as 16, 20, or 24 frames, with temporal modules ensuring consistency across these frames. To generate longer videos while preserving overall consistency, many methods employ the "overlapping video segments" technique. This approach involves segmenting the target video into segments of length $x$ and creating overlaps between consecutive segments. During each denoising process, the overlapping regions between video segments are averaged to maintain consistency, ensuring that each segment aligns with its overlapping regions with others, thereby sustaining overall video coherence.

However, this method has an inherent limitation: the overlapping frames between video segments remain static during each denoising iteration. The frames in the middle of each segment that are not included in the overlap do not directly influence the generation of adjacent segments; they merely exert an indirect effect on other frames through "consistency propagation". This indirect guidance is less effective than the direct guidance provided by the temporal module. The direct influence range for these frames is confined to the generation frame count $x$, whereas the overlapping sections have a direct influence range of $2x$.

To fully harness the "guidance" potential of each frame, we optimize the conventional "overlapping video segments" method by extending the direct guidance range of the middle frames to $2x$. Specifically, a cyclic overlap strategy is adopted instead of using fixed overlapping frames. Following each denoising process for all frames, the frame index of each video segment is incremented by one, keeping them within the proper range. In the subsequent denoising process, the indexes of the overlapping parts between video segments are also incremented by one, resulting in the overlap shifting to the right by one frame. Since the middle frames, excluding the overlap, are typically fewer than the denoising steps, each middle frame participates in the overlap for at least one denoising process, thereby extending its direct influence range from $x$ to $2x$.

As shown in the Appendix, the cyclic overlap generation method enhances video consistency.

## 5 EXPERIMENTS

### 5.1 IMPLEMENTATION DETAILS

We utilized the public VFHQ (Xie et al., 2022) dataset for our experiments, which comprises over 16,000 high-fidelity clips from a variety of interview scenes. The training procedure was divided into two stages. In the initial stage, the Temporal layer was excluded, and both the reference network and the denoising network were initialized using SD1.5. The pose encoder was initialized with Gaussian noise, and a zero convolution was applied at its final layer. During training, two different frames from the same video were sampled to train the reference network, denoising network, and

pose encoder, with the objective of enhancing the model's single-image generation capability. In the second stage, the Temporal layer was incorporated, initialized with AnimateDiff, similar to AnimateAnyone. During this phase, the weights of all other components were frozen, and only the Temporal layer was trained to improve the temporal consistency of video generation.

Throughout both stages, the weights of the VAE encoder, decoder, and CLIP image encoder were kept frozen. We employed the Adam optimizer to train the entire model on an NVIDIA 4090 GPU. In the first stage, the batch size was set to 1, and the model was fine-tuned for 120,000 steps. In the second stage, the batch size remained at 1, with the video length set to 24 frames, and the model was fine-tuned for an additional 30,000 steps. The learning rate for both stages was maintained at 1e-5.

## 5.2 EVALUATION AND COMPARISON

### 5.2.1 QUALITATIVE COMPARISON

We performed a qualitative comparison between our method and several recently popular diffusion-based portrait animation models, including AnimateAnyone (Hu, 2024), MagicDance (Chang et al., 2023), X-Portrait (Xie et al., 2024), and FollowYourEmoji (Ma et al., 2024), as depicted in the Figure 2. These models were chosen as benchmarks due to their distinguished performance in the domain of portrait animation.

In scenarios involving subtle expressions or cross-id cases with significant differences, these methods either fail to accurately capture the nuanced expressions or are unable to preserve the identity features, leading to noticeable identity shifts. In contrast, our method not only transfers subtle expressions with high precision but also effectively maintains the original identity features. Additional comparison images can be found in the appendix.

### 5.2.2 QUANTITATIVE COMPARISON

To comprehensively assess the method, a quantitative comparison was conducted with state-of-the-art diffusion-based portrait animation methods, as presented in Table 1. The evaluation metrics employed include: (1) Self Reenactment: SSIM (Wang et al., 2004), PSNR (Hore & Ziou, 2010), and LPIPS (Zhang et al., 2018) were computed to assess image-level quality. In the test set, the first frame of each video is used as the reference image, while all subsequent frames serve as the target images and simultaneously as the ground truth. (2) Cross Reenactment: Identity similarity, expression similarity, and image quality were selected as metrics to evaluate cross-id generation performance. ArcFace (Deng et al., 2019a) was employed to extract identity vectors for each generated video frame and the reference image, and their cosine similarity was calculated. For expression similarity, Deep3DRecon (Deng et al., 2019b) was used to extract parameters such as rotation, translation, and expression from the generated and target video frames, performing a frame-by-frame L1 similarity calculation. Image quality was assessed using QAlign (Wu et al., 2023).

Table 1: Quantitative comparison with state-of-the-art diffusion-based portrait animation methods.

| Method | Self Reenactment | | | Cross Reenactment | | |
| --- | --- | --- | --- | --- | --- | --- |
| | SSIM ↑ | PSNR ↑ | LPIPS ↓ | Identity ↑ | Expression ↓ | Image Quality ↑ |
| MagicDance | 0.594 | 16.368 | 0.249 | 0.288 | 18.629 | 3.473 |
| AnimateAnyone | 0.653 | 18.72 | 0.186 | 0.290 | 19.059 | 4.089 |
| FollowYourEmoji | 0.677 | 20.274 | 0.154 | 0.451 | 20.301 | 4.149 |
| X-Portrait | 0.629 | 18.729 | 0.211 | **0.504** | 18.626 | 3.977 |
| Ours | **0.692** | **20.304** | **0.150** | 0.493 | **16.953** | **4.318** |

As shown in the Table 1, in the same-id scenario, our method outperforms all others. In the cross-id scenario, FollowYourEmoji and X-Portrait exhibit strong performance in identity preservation, owing to their cross-id training strategies. MagicDance and X-Portrait excel in capturing and transferring expressions, as they utilize processed RGB images as reference actions, thereby retaining more pose and expression information. AnimateAnyone and FollowYourEmoji surpass others in

terms of image quality. However, our method ranks first across most of the metrics, demonstrating that our generated results are superior in terms of identity preservation, expression transfer, and image quality. Notably, there is a significant improvement in the expression metric, with an 8% increase over the previous state-of-the-art methods.

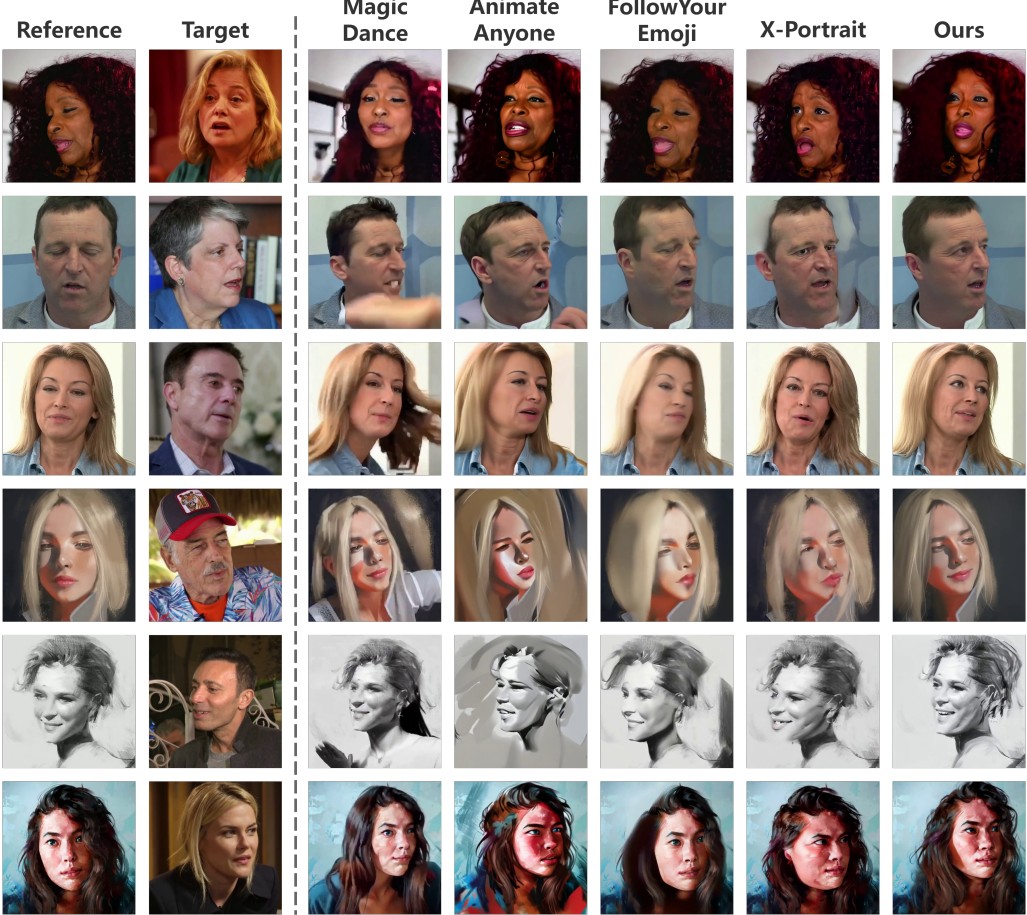

Figure 2: **Qualitative result compared with SOTA methods.** Given a reference portrait image and target portrait images, our approach demonstrates superior performance in accurately capturing fine facial expressions while maintaining the original identity of the characters, outperforming previous methods.

## 5.3 ABLATION STUDY

We performed ablation studies to evaluate the efficacy of the random scaling of facial parsing maps, the foreground-background separation mechanism, and the optimized long-video generation strategy. Visual results and additional experiments are presented in the Appendix. The quantitative results, as presented in the Table 2, further substantiate the efficacy of our optimized strategy.

First, both partial and overall scaling were eliminated from the facial parsing maps, and the outcomes were compared with those achieved through complete scaling. Omitting the scaling process results in substantial identity leakage, with the generated output failing to preserve the identity features of the reference image and incorporating excessive identity elements from the target image, such as facial shape and nose size.

To assess the influence of the foreground-background separation on the generated results, the complete image was encoded using CLIP and input into the ReferenceNet and DenoisingUNet for train-

ing. The results demonstrate that using the complete image yields inferior outcomes in both identity and background preservation compared to the foreground-background separation method, leading to increased identity shifts and background loss.

Table 2: Quantitative results of ablation study.

| Method | Self Reenactment | | | Cross Reenactment | | |
|---|---|---|---|---|---|---|
| | SSIM ↑ | PSNR ↑ | LPIPS ↓ | Identity ↑ | Expression ↓ | Quality ↑ |
| w/o random scaling | 0.682 | 20.025 | 0.157 | 0.305 | 17.661 | 4.243 |
| w/o separation mechanism | 0.652 | 19.117 | 0.183 | 0.489 | 18.209 | 4.225 |
| w/o cyclic overlap | 0.657 | 19.289 | 0.178 | 0.488 | 18.090 | 4.242 |
| full model | **0.692** | **20.304** | **0.150** | **0.493** | **16.953** | **4.318** |

Lastly, we conducted a comparison between the conventional long-video generation method and our optimized approach. The findings indicate that, over extended durations, our model consistently exhibits superior frame-to-frame consistency while effectively preserving the identity features of the reference character.

## 5.4 LIMITATIONS AND FUTURE WORK

In the future, we intend to incorporate finer facial details to further enhance the model's capabilities. The current facial parsing maps are limited in their ability to simulate tongue movements and inadequately capture expressions such as frowning, which exhibit low correlation with the shape and position of facial components. To address these limitations, we will integrate diverse facial representation methods. Moreover, we plan to explore more advanced temporal attention mechanisms and multi-segment generation strategies to further improve frame-to-frame consistency.

## 6 CONCLUSION

We have introduced Vividportraits, an innovative portrait animation framework grounded in diffusion models. Our approach effectively disentangles identity, motion, and background information during training, enabling precise transfer of facial expressions and movements while preserving robust identity features and background consistency. Furthermore, we presented an optimized long-video generation strategy that ensures each video frame receives sufficient attention. Experimental results indicate that our model excels in portrait animation tasks, delivering impressive outcomes. Looking forward, our framework holds the potential for application and extension to more complex animation tasks.

## REPRODUCIBILITY

Our code is included in the supplementary materials, comprising the complete data processing, training, and inference code. This facilitates the reproducibility of our work.

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

# A APPENDIX

## A.1 ABLATION STUDY OF RANDOM SCALING OF FACIAL PARSING MAPS

We omitted the random scaling process and retrained the model. The results, as shown in the first row of Figure 3, indicate that when there is a significant discrepancy in facial shape between the target and reference characters, the generated output loses many original identity features. This suggests that the model has learned excessive identity characteristics from the target parsing map, rather than focusing solely on expression and motion features.

The model was trained without the incorporation of eye landmarks. The second row of Figure 3 illustrates that, under these conditions, the model fails to capture information related to eyeball position, leading to the generation of rigid or vacant facial expressions.

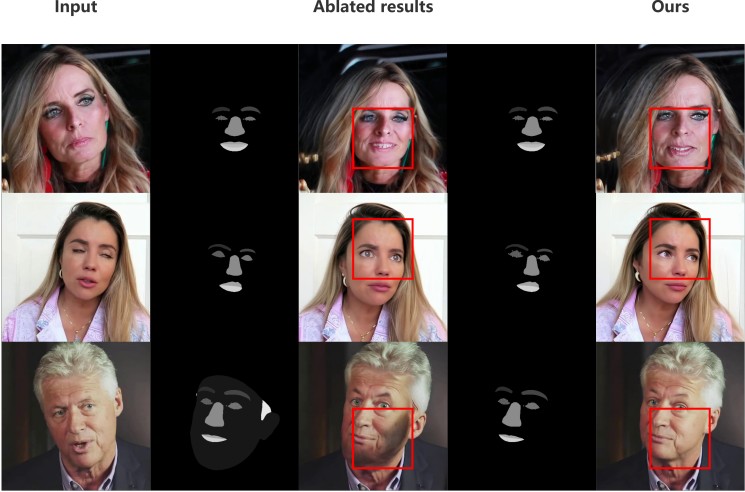

Figure 3: Ablation study of random scaling of facial parsing maps. The first row indicates that training without random scaling tends to extract identity information from the target portrait's facial parsing map. The second row shows that training without incorporating eye landmarks results in poor interpretation of eyeball position information. The third row demonstrates that retaining facial contours during training leads to more noticeable artificial artifacts.

The third row of Figure 3 indicates that when the model is trained using facial parsing maps that include contours, it generates facial shapes based on these contours, consequently modifying the identity of the reference character.

## A.2 ABLATION STUDY OF FOREGROUND-BACKGROUND SEPARATION MECHANISM

The entire reference image was processed using CLIP and then passed to ReferenceNet and DenoisingUNet, resulting in the retraining of the model. The first row of Figure 4 indicates that, under

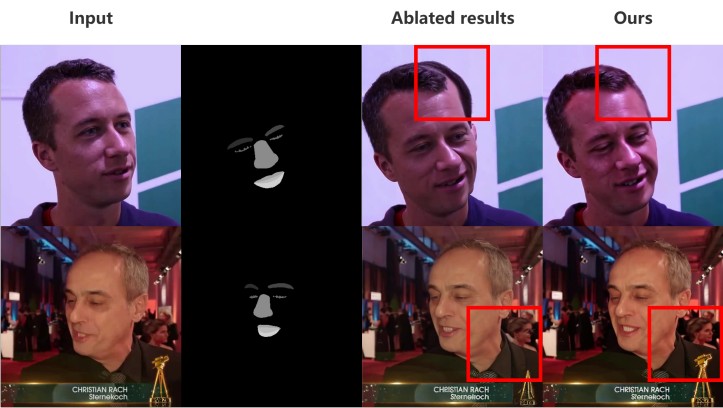

Figure 4: Ablation study of foreground-background separation mechanism. Our foreground-background separation mechanism enables the model to effectively preserve both identity features and background details.

this training approach, the model has difficulty effectively preserving identity information, while the second row illustrates that background information is more susceptible to loss.

### A.3 ABLATION STUDY OF OPTIMIZED LONG-VIDEO GENERATION STRATEGY

During inference, we generated results using fixed overlapping regions and compared them with the outcomes from our cyclic overlap generation method. We compared the frames generated after a longer duration (10 seconds) with the first frame, with the results displayed in Figure 5. The cyclic overlap generation method demonstrates superior performance in preserving the features of the first frame.

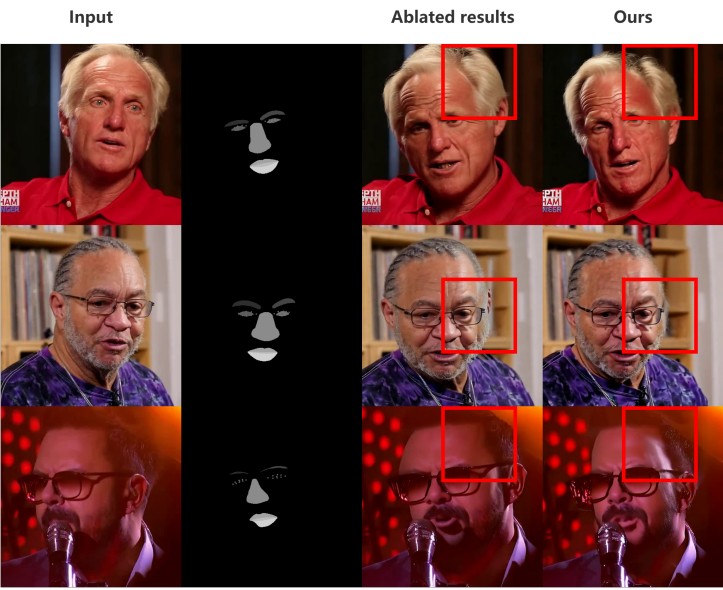

Figure 5: Ablation study of optimized long-video generation strategy. Over prolonged periods, optimized strategy consistently achieves better frame-to-frame consistency.

