# OpenReview forum: "Vividportraits: Face Parsing Guided Portrait Animation"
_ICLR.cc/2025/Conference — ICLR 2025 Conference Withdrawn Submission_

### Official Review · Reviewer_g3sa · 2024-10-29

**Soundness:** 2
**Presentation:** 3
**Contribution:** 2
**Rating:** 3
**Confidence:** 5

**Summary:**

This paper presents Vividportraits, a portrait animation method based on diffusion models. Its key innovation is the use of facial parsing maps for motion representation. Several technical enhancements are introduced, including random scaling of facial parsing maps to prevent identity leakage, foreground-background separation for improved background handling, and the implementation of a cyclic overlap technique to enhance long-term generation quality.

**Strengths:**

This paper propose to use facial parsing maps instead of facial landmarks as motion representation, and propose several technical improvements to enhance identity preserving and long-term generation quality.

**Weaknesses:**

1.  I don't understand why facial parsing maps are better than landmarks. The paper lacks a fair ablation study to justify this choice.
2. The innovation is quite limited, as the reference-net structure is already well-established in SD-based animation.  The proposed improvements are more like training/inference tricks to me. Additionally, the overall quality is not impressive.
3. Missing discussion and comparison with some recent works, e.g.
LivePortrait: Efficient Portrait Animation with Stitching and Retargeting Control.

**Questions:**

See Weaknesses.

**Details Of Ethics Concerns:**

No ethics concerns.

---

### Official Review · Reviewer_dYd3 · 2024-11-02

**Soundness:** 2
**Presentation:** 3
**Contribution:** 2
**Rating:** 3
**Confidence:** 4

**Summary:**

This paper proposes an SD-based face reenactment method that uses a face parsing map as the driving condition. To reduce the correlation between face parsing and facial shape, the authors implemented a strategy with a random scale. They also proposed foreground-background separation and cyclic overlap sampling strategies to enhance visual quality and the results of long video generation. The paper claims to have achieved state-of-the-art results, but I hold a different opinion, which is detailed in the weaknesses section.

**Strengths:**

1. The authors propose a new portrait-driven modality: face parsing.
2. The figures in the paper makes the proposed method easy-to-follow, and the authors provide detailed code to facilitate reproducibility.
3. The authors designed ablation studies to verify the effectiveness of the designed modules.

**Weaknesses:**

1. I have some doubts about the rationale of using a parsing map for portrait animation. While parsing maps can capture more details than keypoints in some cases, face parsing faces numerous challenges beyond those mentioned in the paper. The inherent robustness of face parsing is weak, making it susceptible to performance impacts from occlusions, such as partial obstructions of the eyes or corners of the mouth. Additionally, face parsing, as the paper states, cannot capture eyelid or mouth movements. There are also denser keypoint sets, such as the 203 keypoints from  InsightFace[[1]](https://github.com/deepinsight/insightface) or 478 keypoints from MediaPipe[[2]](https://github.com/google-ai-edge/mediapipe/blob/master/docs/solutions/face_mesh.md).
2. The proposed method does not significantly differ from other related methods, such as AnimateAnyone or Halo; technically, it is incremental.
3. My another concern is that the videos presented in the paper, which use face parsing as motion guidance, exhibit considerable jitter and inter-frame inconsistency, especially in the instability of hair.
4. The paper mentions that using a random scale for face parsing and removing the parsing for the face area can somewhat alleviate the coupling between the parsing map and facial shape. However, other areas, such as the shape and size of the nose, still remain correlated with the person's shape.
5. I noticed that previous GAN-based and motion flow-based face reenactment methods, such as Face-vid2vid[[3]](https://nvlabs.github.io/face-vid2vid/) and DA-GAN[[4]](https://github.com/harlanhong/CVPR2022-DaGAN), etc., have also achieved high-quality face reconstruction and reenactment results. However, the paper does not include relevant comparisons.
6. The driving results shown in Fig 2 of the paper indicate that the head pose and expressions differ from the driving images, and in some cases, they perform worse than other comparison methods, such as the results in the last two rows.

**Questions:**

1. Lines 306-308 are not very easy to understand; a more intuitive explanation is needed.
2. Why do the identities in Table 1 show poor performance in cross reenactment? Is it affected by the shape of the driving person?

---

### Official Review · Reviewer_pfFD · 2024-11-03

**Soundness:** 2
**Presentation:** 3
**Contribution:** 1
**Rating:** 3
**Confidence:** 4

**Summary:**

This paper proposes a facial reanimation using a diffusion model. The overall architecture is very similar to that of AnimateAnyone with the driving signal being face segmentation maps instead of the guiding pose. A ReferenceNet is used to extract features from a single reference frame, which are used to conditioning a diffusion model. The same diffusion is also conditioned to generate the target head pose and facial expression using a pose guider network. Finally, a cyclic overlapping strategy is used to generate temporally consistent videos.
The method was compared quantitatively and qualitatively with prior art on the VFHQ dataset.

**Strengths:**

1) Each component of the method is well ablated
2) Paper is well written
3) Method performs better than prior art both quantitatively and qualitatively

**Weaknesses:**

1) While the method performs well quantitative and qualitatively, I believe it is only a small modification of Animate Anyone. AnimateAnyone showed that a ReferenceNet + PoseGuided diffusion model can be used to animate humans (bodies and or faces). This paper does not further our understanding of the problem or solve any superset of the problem (such as relighting or hair animation to name a few). The way I see it, this paper takes a method and makes it more robust using stronger augmentations and sampling tricks. I do not believe, in its current form, it warrants an acceptance

**Questions:**

One possibility worth investigating is the efficacy of each representation of a face for reanimation. The proposed method uses face parsing maps, but those can be replaced with other representations such as landmarks, or a mesh, or reference images themselves. It would be very interesting to see how each one of them performs and which ones is the best, if at all there’s a clear winner. I believe such an analysis of representations would make the paper significantly stronger as it would contribute towards an understanding of the problem.

---

### Official Review · Reviewer_nGSn · 2024-11-05

**Soundness:** 3
**Presentation:** 3
**Contribution:** 3
**Rating:** 8
**Confidence:** 4

**Summary:**

This paper proposes a novel portrait animation method based on diffusion models that improves control over facial expressions and poses while maintaining character identity. It introduces a new region-based motion representation using facial parsing maps for accurate expression capture, a cross-identity training approach that prevents identity mixing, and a foreground-background separation to reduce noise and enhance feature extraction. Furthermore, the paper proposes a "cyclic overlap" technique for improved temporal consistency in long-video generation.

**Strengths:**

+ The paper is very well-written.

+ The proposed framework is sufficiently novel.

+ The experimental evaluation is thorough and the proposed method is compared with several recent SOTA methods. The visualizations in the figures as well as the presented metrics clearly show that the proposed Vividportraits outperforms the previous SOTA methods included in the experimental comparisons

+ The experimental evaluation includes also an ablation study that provides quantifiable evidence that all main modules of the pipeline (random scaling, separation mechanism and cyclic overlap) have a positive effect in the method's performance. Furthermore, the appendix includes some visualizations that show that this is reflected in the perceptual visual quality of the generated frames.

**Weaknesses:**

- The isolated individual frames, as shown in the figures of the paper, are of high quality. However, observing the videos in the supp. material, one can see that the temporal consistency of the generated videos has some notable artifacts. See for example the neck area in 001.mp4 and 002.mp4, as well as the hair and right ear's earring in 001.mp4.

- Furthermore, these videos do not include any comparisons with previous SOTA methods, which is insufficient, since it is not possible to qualitatively compare the temporal consistency of the proposed model as compared to the previous ones. It could be the case that previous SOTA methods exhibit better temporal consistency and that this comparison has been omitted from the submission.

**Questions:**

I'd like to read the authors' view regarding my criticism on the limitations regarding temporal consistency, as well as the lack of comparisons with previous SOTA methods in terms of generated videos.

**Details Of Ethics Concerns:**

This method can be used to generate highly-realistic deepfake videos, without the consent of the depicted individual. There are several ethical issues that need consideration.

---

### Note · Authors · 2024-11-15

I have read and agree with the venue's withdrawal policy on behalf of myself and my co-authors.